# Muscle Activity and Co-Activation of Gait Cycle during Walking in Water and on Land in People with Spastic Cerebral Palsy

**DOI:** 10.3390/ijerph20031854

**Published:** 2023-01-19

**Authors:** Pariyaporn Phothirook, Sugalya Amatachaya, Punnee Peungsuwan

**Affiliations:** 1Research Center in Back, Neck, Other Joint Pain and Human Performance (BNOJPH), Khon Kaen University, Khon Kaen 40002, Thailand; 2School of Physical Therapy, Faculty of Associated Medical Sciences, Khon Kaen University, Khon Kaen 40002, Thailand

**Keywords:** electromyography, water gait, cerebral palsy, muscle co-activation, muscle activity

## Abstract

Background: The purpose of this study was to investigate the differences in the muscle activity and co-activation index (CoA) of the rectus femoris (RF), biceps femoris (BF), gastrocnemius medialis (GM,) and tibialis anterior (TA) during walking on land and in water in healthy adolescents compared with those with spastic diplegia cerebral palsy (CP) adolescents. Methods: Four healthy individuals (median; age: 14 years, height: 1.57 cm, BMI: 16.58 kg/m^2^) and nine CP individuals (median; age: 15 years, height: 1.42 cm, BMI: 17.82 kg/m^2^) participated in this study and performed three walking trials under both conditions. An electromyography (EMG) collection was recorded with a wireless system Cometa miniwave infinity waterproof device, and the signals were collected using customized software named EMG and Motion Tools, Inc. software version 7 (Cometa slr, Milan, Italy) and was synchronized with an underwater VDO camera. Results: A significant decrease in the muscle activity of all muscles and CoA of RF/BF muscles, but an increase in TA/GM was observed within the CP group while walking in water during the stance phase. Between groups, there was a lower CoA of RF/BF and a greater CoA of TA/GM during the stance phase while walking in water and on land in the CP group. A non-significant difference was observed within the healthy group. Conclusion: Walking in water can decrease muscle activity in lower limbs and co-activation of thigh muscles in people with spastic CP, whereas CoA muscles around ankle joints increased to stabilize foot weight acceptance.

## 1. Introduction

Cerebral palsy (CP) is described as a group of motor disorders primarily affecting the development of movement and posture, causing a limitation of activity [1]. The motor disorders of individuals with CP present a complex that relates to muscle spasticity, muscle weakness, and loss of selective motor control. This leads to muscle contractures and bony deformities, which cause deviations in the gait. Walking is essential for activities of daily living and social participation; therefore, it is often considered one of the most important activities in daily life. People with spastic CP have increased muscle contractions and over co-activation between agonist and antagonist muscles that affect an abnormality in the gait cycle.

Muscle activity during a gait cycle has been reported in a few previous studies for people with spastic CP, but almost all those were interested in studying walking on land [2,3,4,5]. A previous study investigated muscle activity during walking in water and dry land in healthy children and children with CP, and they were suggested to have a better understanding of the extent of muscular activation [6]. Muscle co-activation between agonist and antagonist muscles is concurrently activated excessively during locomotion and is commonly a characteristic of people with spastic CP; notably, overaction in flexion of knees in the CP, which may have been brought about by co-contraction of the hamstrings and rectus femoris [5,7]. Thus, the study of changes in muscle co-activation during walking is very important for the clarification of muscular function in the gait cycle.

Hydrotherapy is an alternative intervention to use in the clinical rehabilitation of individuals with neurological disorders. Many physicians and physical therapists who have recommended hydrotherapy have reported positive benefits in people with CP [8]; however, there is no better understanding of muscle activity and co-activation during the gait cycle in CP on the effect of body immersion or walking in water. The present study may lead to an understanding to make a clinical decision for walking-based hydrotherapy in people with CP. The purpose of this study was, therefore, to investigate the surface electromyography (sEMG) activity and co-activation in the pairs of flexor and extensor of knee and ankle muscles; the rectus femoris (RF), biceps femoris (BF), gastrocnemius medialis (GM) and tibialis anterior (TA), and the spatiotemporal parameters in people with spastic diplegia CP, and to compare these findings with healthy people during walking in water and on land.

## 2. Materials and Methods

### 2.1. Subjects

The research design was a case-controlled study that compared spastic CP subjects with healthy subjects to examine their muscle activity and co-activation in different gaits in water and on land.

The sample size was calculated referred based on a previous study outcome [6] using g-power software to provide α = 0.05, 80% statistical power to sample size determination.

The study sample required at least nine subjects per group, however, four healthy individuals were used for data collection due to the COVID-19 pandemic situation. 

Nine subjects with CP aged 10–18 years were recruited from a special school, Srisangwan Khon Kaen, Thailand, who were diagnosed with a type of spastic diplegia/ diparesis and were classified by GMFCS-E&R levels of I or II. Inclusion criteria consisted of being able to perform walking over a 5-m distance without assistive devices but having good communication capability and could follow the assessor’s instructions. Subjects would be excluded if found to have other problems such as seizures, unfamiliar water surroundings, instability in walking, and/or uncooperativeness, visual or hearing impairment, musculoskeletal surgery, or medical treatment, e.g., botulinum toxin or baclofen) within the previous 6 months. All individuals with CP were evaluated for their leg muscle tone on both sides of the hip flexor/extensor, hip internal rotator/external rotator, hip adductor/abductor, knee flexor/extensor and ankle dorsiflexor/plantar flexor using the Modified Ashworth Scale (MAS) prior to trial. Four healthy adolescents were recruited if they had demographics of age, weight, and height similar to subjects with CP, and they would be invited to participate. The healthy individuals studied at the elementary school and had high academic levels. All the participants and their parents read and signed an informed consent document that was approved by the Institutional Ethics Committee for Human Research at Khon Kaen University, Thailand (HE 622282).

### 2.2. Experimental Procedure

Electromyography (EMG) is widely utilized in assessment and biofeedback therapy for children with CP to improve lower limb motor function, gait speed, neuromuscular control, and motor coordination [9]. The surface EMG (sEMG) was utilized to measure muscle activity in this study.

#### 2.2.1. EMG Assessment

Surface EMG activity of the rectus femoris (RF), biceps femoris (BF), gastrocnemius medialis (GM), and tibialis anterior (TA) of both lower extremities were recorded during the gait trials on land and in water. The sensor EMG electrodes were positioned according to the most validated recommendations for surface electromyography according to the Non-Invasive Assessment of Muscles (SENIAM), in which the RF electrode was placed halfway between the line from the anterior superior iliac spine to the superior part of the patella. The BF electrode was placed halfway on the line from the ischial tuberosity and the lateral epicondyle of the tibia. The GM electrode was placed on the most prominent bulge of the muscle. The TA electrode was placed on one-third of the line between the fibular tip and the tip of the medial malleolus [10]. The electrodes were connected to an EMG data collection system with the wireless apparatus of a Cometa miniwave infinity waterproof device and the signals were collected using customized software named EMG and Motion Tools, Inc. software version 7 (Cometa slr, Milan, Italy). Prior to the placement of the electrodes, the skin was cleaned by scrubbing and wiped with 70% alcohol. The Ag/AgCl bipolar surface adhesive electrodes (H124SG, Covidien Inc., Minneapolis, MN, USA) were 2.4 cm in diameter, and they were placed at 1 cm interelectrode distances. During EMG recording in water, the electrodes and the sensors were covered using a transparent and waterproof adhesive film (20 cm × 20 cm) to prevent movement and loosening of the sensors during the walking process.

For EMG recordings, each participant was instructed to walk barefoot at a self-selected comfortable speed on land and in water for a distance of 5 m. Body immersion in the pool was set at the participant’s umbilicus level, and the temperature was controlled to be 32–33 °C. Water level can reduce half a participant’s body weight while resulting in fewer alterations of posture and less turbulence at the umbilicus level compared with deeper immersion [11]. All participant walking was videotaped with synchronization to the EMG underwater during walking at self-selected comfortable speeds and they performed a preliminary test at a preferred speed to familiarize themselves with the surroundings and the experimental device. They were asked to walk a total of three trials with a 1-min rest between trials in both environmental conditions.

Data analysis and processing were performed using the raw EMG signal fully rectified offline and calculated as a high-pass filtered cut-off frequency of 20 Hz and a low-pass filtered cut-off frequency of 400 Hz. The EMG amplitude was calculated using the integrated EMG (iEMG) algorithm. All EMG records were explored from video recordings and the data were selected from similar cycles in the six gait cycles of three gait trials, with the average iEMG muscle activity being calculated for each participant. The iEMG data in each phase was presented as the median iEMG due to non-normal data distribution.

A gait cycle was divided into eight sub-phases and started at the instant when one foot touched the floor and stopped when the same foot came into contact for the next step, based on the literature [12]. Each sub-phase consisted of (1) initial contact, (2) loading response, (3) mid-stance, (4) terminal stance, (5) pre-swing, (6) initial swing, (7) mid-swing, and (8) terminal swing (Figure 1). These were identified using visuals associated with the synchronized EMG video recording.

#### 2.2.2. Muscle Co-Activation

Co-activation has been quantified using an index typically based on direct recording of muscle activation from both muscles within an agonist–antagonist pair called the “co-activation index or CoA”. The iEMG activity was calculated from the eight sub-phases across two muscle pairs as RF/BF, BF/RF, TA/GM, and GM/TA, using the following index [4,13] below:
(1)$$\mathrm{CoA}=\frac{\mathrm{EMG}\;\mathrm{antagonist}}{\mathrm{EMG}\;\mathrm{agonist}}\times 100$$

Defining co-activation, RF was defined as the agonist during initial contact, loading response, mid stance for limiting the magnitude of flexion occurring as the foot strikes the ground, mid-stance and terminal stance to the initiating knee extension, and pre-swing to control for knee flexion. BF was defined as the agonist during initial-, mid-, and terminal-swing in initiating knee flexion, preparing for foot clearance and controlling for knee extension and decelerating the swinging leg. TA was defined as an agonist during the weight acceptance phase (initial contract and loading) in working to control the lowering of the foot and during initial-, mid-, and terminal-swing in lifting the foot from the ground and ensuring foot clearance. GM was defined as the agonist during mid-stance, terminal stance, and pre-swing, where they are the main contributors to stabilization, control for ankle dorsiflexion, and preparation for foot off as presets, following [4].

#### 2.2.3. Spatiotemporal Data

The spatiotemporal parameters were calculated: walking speed, cadence, and gait cycle time which was simultaneous with the EMG and videotape recording. The duration of the stride of the stance phase and swing phase periods were averaged as percentages [14].

### 2.3. Statistical Analysis

Descriptive statistics were applied to explain the demographics and characteristics of the subjects and the findings of the study. All data analyses were performed using SPSS for Windows version 17.0 (SPSS Inc., Chicago, IL, USA). The data were checked for normal distribution with the Shapiro–Wilk test and histograms. When the data did not show normal distribution, a non-parametric statistical analysis was also applied and the Wilcoxon signed-rank test and Mann–Whitney U test were used in comparisons during gait cycles between land and in water and those between groups. The power of the test was 0.8. The significance level was set at an alpha lower than 0.05 (*p* < 0.05).

## 3. Results

### 3.1. Participant Characteristics

Participant characteristics are shown in Table 1. Nine people with spastic diplegic CP aged 15 years, median (Q1, 3: 13, 16; range: 12 to 17 years). Eight males and one female characterized the GMFCs levels I (n = 7) and II (n = 2). Four healthy subjects were aged 14 years, median (Q1, 3: 12.75, 15.25; range: 12 to 16 years). Heights and weights were similar.

### 3.2. Spatiotemporal Parameters

Spatiotemporal parameters during a gait cycle found a significant difference are shown in Table 2. Within-group differences, an increase in the gait cycle time, and the percent of the stance phase were significantly different, while a reduction in the walking speed and the percent of the swing phase during walking in water were compared on land in the CP group. No significant differences occurred in all parameters when compared during walking in water and on land in the healthy group.

### 3.3. EMG Activities during Gait Cycle

Comparing the median iEMG activities during the gait cycle while the healthy (n = 4) and CP (n = 9) individuals walked on land and in water are shown in Figure 2, Figure 3, Figure 4 and Figure 5. A significant median difference was compared between two conditions or/and groups were presented as scores of the median differences and *p*-values (in parenthesis), which was mentioned in the results.

#### 3.3.1. Comparing in Water and on Land in Healthy Subjects

The EMG activity presented as lower in all muscles when walking in water compared with on land. The study did not find a significant difference (Figure 2).

#### 3.3.2. Comparing Subjects with CP in Water and on Land

Figure 3 shows a comparison of the EMG activity during walking in two conditions. The activity of all muscles was significantly higher level when individuals with CP walked on land compared to walking in water. Those are significant median differences in the RF (Lt.: 2.37, *p* = 0.002) during the mid-stance (Figure 3A) and the BF during the mid-stance (Rt.: 2.00, *p* = 0.04), the loading response (Lt.: 1.11, *p* = 0.01) and both sides during the initial swing (Rt.: 1.73, *p* = 0.01; Lt.: 1.63, *p* = 0.008) (Figure 3B). The TA during terminal swing (Rt.: 0.50, *p* = 0.02), the GM during loading response (Rt.: 2.31, *p* = 0.01), and the mid stance (Rt.: 1.10 *p* = 0.04), which are presented in Figure 3C,D.

#### 3.3.3. Comparing between Groups on Land

The general observation graphs showed a higher level of all the muscle activities in the CP group compared with the healthy group, and significant median differences in the RF, BF, GM, and TA were found in various sub-phases (Figure 4). Muscles controlling the hip and knee parts are represented by the RF and the BF, which demonstrated a greater activity from an initial contract to initial swing phases (Figure 4A,B). Parts of ankle-controlled muscle are the TA and GM, which showed a greater activity in two stance sub-phases (loading response and pre-swing) and three swing sub-phases (initial, mid, and terminal swing) (Figure 4C,D).

#### 3.3.4. Comparing between Groups in Water

An overview of the RF, BF, TA, and GM muscle activities was significantly greater in the CP group than in the healthy group during the gait cycle in all of the sub-phases (*p* = 0.05 and 0.01), shown in Figure 5. Regarding all of the data can indicate that significant median differences were found in the RF of all sub-phases (Figure 5A) and the BF in a few sub-phases (Figure 5B) during weight acceptance, while The TA was found during weight acceptance and swing phases (Figure 5C), but the GM during weight acceptance (initial contract, loading response, and pre-swing) (Figure 5D).

### 3.4. Muscle Co-Activation during Gait Cycle

#### 3.4.1. Comparing in Water and on Land in Healthy Subjects

No significant median differences in the right and left sides of the RF-BF CoA and the TA-GM CoA of all the sub-phases compared between walking on land and in water (Figure 6). The data seem to show that the RF-BF CoA showed greater when walking on land during terminal stance and pre-swing phases but less during the initial contract.

#### 3.4.2. Comparing in Water and on Land in Subjects with Cerebral Palsy

A significantly greater BF/RF CoA on the left side during loading response (190.03, *p* = 0.03) when individuals walked on land than walking in water are shown in Figure 7B. In comparison, the CoA of TA/GM were significantly greater during the mid-stance (Rt.: 55.29, *p* = 0.03) (Figure 7C) and terminal-stance (Lt.: 94.56, *p* = 0.008) (Figure 7D) while individuals walked in water compared with walking on land.

#### 3.4.3. Comparing between Groups on Land

The CoA of BF/RF were greater in the healthy group during the initial contract (Rt.: 103.17, *p* = 0.014) and pre-swing (Rt.: 158.52, *p* = 0.005; Lt.: 186.87, *p* = 0.031) than in the CP group (Figure 8A,B). In contrast, the CoA of GM/TA were significantly rather than in the CP group during the loading response (Rt.: 85.39, *p* = 0.009; Lt.: 56.04, *p* = 0.045) and the CoA of TA/GM during terminal swing (Lt.: 24.26, *p* = 0.045) those shown in Figure 8C,D.

#### 3.4.4. Comparing between Groups in Water

In healthy individuals, the CoA of the right BF/RF shows a significantly greater median during an initial contract (Rt.: 150.38, *p* = 0.005) (Figure 9A). The CP group shows the CoA of GM/TA were significantly greater during the mid-swing (Lt.: 55.52, *p* = 0.021) and the loading response (Lt.: 51.48, *p* = 0.045) of those shown in Figure 9D.

## 4. Discussion

This study presents the first article to investigate the EMG activity and co-activation between concurrent agonist and antagonist in both lower extremities to analyze each gait sub-phase and those compared the healthy and CP individuals during walking on land and in water. This study found non-significant differences in the EMG activity of RF, BF, GM, and TA muscles in each gait sub-phase in the healthy group in comparing walking between land and in water. Within the CP group, there was significantly decreased EMG activity of the four muscles in a few sub-phases of the water gait when compared to land, there was significantly greater EMG activities of all muscles in various gait sub-phases compared with the healthy individuals in both conditions.

There were no significant differences in muscle activity when walking on land and in water for the healthy group. This was similar to a previous study that focused only on the RF activation [6]. Masumoto’s study showed a significant decrease in RF, BF, TA, gastrocnemius, vastus medialis, and gluteus medias muscle activity when walking in water in healthy humans [15]. Although the present study did not find a significant difference in muscle activity, there tended to be a decrease while walking in water compared to land. A previous study suggested that immersion reduces the activity of the extensor muscles of the lower limbs [16], while Nilsson et al. (1985) showed that the resistance of the water to forward movement increases such activity [17].

One of the primary purposes of the present study was to emphasize the muscle activity in the CP group while walking in water. The results demonstrated significantly decreased activities in four muscles compared with walking on land in the four sub-phases; loading (RF, BF, GM), mid-stance (BF, GM), initial (BF), and terminal (TA) swing. Similarly, previous findings reported that EMG activity decreased during underwater walking compared to on dry land in elderly women [18] patients with PD [19]. Only the previous Oliveira and colleague’s study to measure similar EMG activity similar to our study reported no significant difference in the RF activity in the CP group.

The effects and properties of water on body movement can be explained in that it is due to the impact of the weightless on the neuromuscular system allowing decreased muscle activity during walking [20]. In addition, the buoyancy force leads to reduced gravitational stress on the musculoskeletal system, which is an unexpected gravity condition that may reduce the muscle spindles and proprioceptive activity [21], and a reduction in normal reaction force on the pool ground that has been demonstrated in water [22]. It is possible that subjects minimized their efforts while walking in water, which would also have resulted in reduced resistance. It is noted that increased resistance to movement caused by hydrostatic pressure and fluid drag forces is expected to slow down the motion and allow a participant to more consciously control their movements [23,24,25]. Those explanations can support an understanding of typical human movement in water.

Individuals with CP present complex and heterogeneous motor disorders that cause gait deviations. Clinical gait analysis is also needed to identify, understand, and contribute to the treatment to utilize hydrotherapy. This study showed an increased RF, BF, and GM activity started at the initial contact to the initial swing periods (approximately 0–80% of a gait cycle) in the CP group when compared with healthy group, which indicates that a period of weight acceptance on the floor may need over reinforcement when walking in water for CP individuals. The results may involve stabilizing in the ankle, knee, and hip joints in CP individuals [26,27]. The distal leg muscles found that TA and GM increased activity during the swing phase might be due to TA’s weakness to attempt control of the ankle joint [28]. Moreover, poor balance and abnormal gait involving over-flexion of the knees in spastic diplegic CP could be co-activated with an increase in the knee extensor during the stance phase and supplementation of the external force of water (i.e., buoyancy, drag, and hydrostatic), thus in order to overcome the resistance, the individuals with CP had to increase their effort. Similar to a previous study in people with Parkinson’s disease (PD), there was an increase in TA activity when walking in water compared with walking on land [19]. Lorencova et al. assumed that when the patients with PD walked in the pool water, with its unstable floor and highest hydrostatic pressure at the ankle joint, it caused contradirection with the ankle dorsiflexion movement. This may have led to a more active TA to overcome the resistance in each gait phase.

Within the groups, non-significant CoA of all muscle pairs in comparing walking on land and in water was found in the healthy group (Figure 6). The individuals with CP showed a significantly increased BF/RF CoA during loading response (double support I) in walking on land compared with walking in water (Figure 7B); this may relate to excessive BF antagonist contraction while walking on land and reduced contraction when walking in water (Figure 3B). The TA/GM CoA significantly increased when walking in water during mid-stance and terminal stance (single support) (Figure 7C,D), likewise, the agonist GM activity was reduced, and the antagonist TA increased (Figure 3C,D). This current study finding may mention that the BF and RF muscles co-activate a few hip and knee joint controls, while the TA and GM muscles need to control more robustly the ankle joint stabilization in the weight acceptance period during walking in water. This may be the reasons that current study opinions identify the cause of buoyancy force, muscle weakness to stabilized joints, and abnormal gait in individuals with spastic CP.

This current study is the first research of comparisons of muscle CoA between a healthy group and the CP group (Figure 8A,B). The results found that CP individuals, when walking on land, had less BF/RF CoA during double support I and II (initial contract and pre-swing), whereas GM/TA (terminal stance, activated plantarflexor) and the TA/GM (loading, activated dorsiflexor) CoA were greater. Even though the muscle CoA seems to be unable to be explicit in comparing the CP group and the healthy group when walked in water, the EMG activity tends to be higher in the CP group.

The gait on land for the spastic CP subjects was found to be the highest TA/GM CoA during the single support of the leg (mid and terminal stances) (Figure 7C,D), which is consistent with previous studies in healthy females [29]. Previous studies proposed that spasticity of the ankle plantarflexors is the cause of increased muscle over-activity (in intensity and duration) during the gait in people with CP [30,31,32] and multiple sclerosis [26], involving increased ankle muscle co-activation in order to stiffen the ankle joint and improve the stability of weight acceptance and the efficiency of weight transfer, which is an adaptive strategy [33,34]. In contrast, the lowering of the RF/BF CoA in water may relate to the decrease in the RF activity during gait in water in the double support (loading) period (Figure 7B).

Moreover, a few internal and external factors impact on the results, such as body immersion in water may contribute to postural control [35], and walking speed is known to influence muscle activity and muscle co-activation [36]. This current study, the self-selected speed of individuals with CP on the ground is obviously faster than in water (Table 2); thus, this may lead to higher muscle co-activation on land during the loading response. An over-activated co-contraction in hip and knee flexors represents an atypical walking characteristic in people with spastic CP, where antagonist co-contraction during extension has been found to be greater in children with CP than typically developing children [37]. In addition, concurrent co-activation of agonists and antagonists found that an increase in ankle dorsiflexion accompanied by a significant decrease in the plantarflexion moment has also been shown in a few previous studies [12,38,39].

A greater muscle, hip, and knee co-activation in the healthy group compared the CP group in the double support I and II periods when walking on land. These resemble previous studies that reported more increased levels of co-activation in healthy adults than adults who had strokes [40] and individuals CP [34,41]. The greater muscle co-activation in healthy individuals during the transition from stance to swing contribute to the increase in joint stability required at higher walking speeds [42], likewise, the present results of healthy individuals showed a faster walking speed than the CP group (Table 2). The muscle ankle co-activation in loading responses were greater in CP than in healthy in both conditions and in terminal stance on land, which was similar to a previous study in Parkinson’s disease (PD) [43]. Ideally, the individuals’ CP with motor impairments is commonly postural instability, and they also tend to have greater co-activation during walking, particularly in water.

Muscle activity and co-activation during walking in subjects with CP could influence their spatiotemporal gait parameters. This study found a significant decrease in walking speed, gait cycle time, and percent of the stance phase when walking in water, but no change in gait cadence. The results agreed with previous studies in healthy human subjects [23,44,45]. Accordingly, the physical properties of water can explain the spatiotemporal gait cycle as follows: The decrease in stance phase time is probably due to the drag forces acting on the body during walking, which may cause the participant’s postural instability; as a result, the leg lifting up motion was comparatively hurried and this might lead to a shorter support time, and a longer swing phase time because of the buoyancy force that influences a more slowly movement of the lower limb when the foot is placed on the pool floor during mid-swing to terminal swing, including the hydrodynamic resistance that contributes to slow down subjects’ movement as well.

The present findings provide valuable information that contributes to the design of water-based exercise programs for rehabilitation. That is, walking in water could decrease muscle activity in spastic lower limbs, which may generate easier stepping and lead to success in walking. In addition, walking in water may promote normalizing muscle tone in preparation for physical functional training; finally, the improved walking ability could enhance daily living for persons with CP. This finding is likely suitable for persons with spastic CP.

There were several limitations of this study. A small sample size in healthy humans could limit the statistical power to expose such a small effect, possibly resulting in a type II error. Because of the study was conducted during the COVID-19 pandemic situation, which caused an incomplete sample number. Nevertheless, the healthy subjects’ demographic characteristics (age, height, and weight) were similar to the spastic CP subjects, and showed no significant difference in spatiotemporal parameters (cadence and gait cycle time). Consideration of the muscle activity (iEMG) variable seems a small variation in the healthy group.

Moreover, non-homologous symptoms in cerebral palsy with spastic diplegia could not be identified as the same. It is difficult to provide general considerations for the gait deviations in CP due to multiple gait deviations coming from different combinations of impairments. The study could not clarify the difference in the level of severity between the right or left sides of the leg, therefore, it was decided to explain all of the outcomes on both leg sides. Finally, the levels of spasticity in the individuals with CP were measured by the Modified Ashworth Scale, with a low level of 1 to 2 scales of each leg muscle, which seemed to be no difference in each leg side.

## 5. Conclusions

The present study demonstrated that EMG activities of the RF, BF, GM, and TA while walking in water decreased in CP individuals with spastic diplegia. Muscle CoA significantly increased in the hip and knee muscle (double support I) for walking on land and an ankle muscle (single support) for walking in water within the CP group. Between groups, the CP group showed a lower hip and knee muscle CoA during double support I and II when walking on land, whereas a greater ankle muscle CoA during double support I and single support when walking on land and in water, but a lower during mid-swing in water. EMG gait analysis on land compared with in water can provide objective identification of gait muscle activations and co-concurrent muscle contraction, especially when walking in water. Further studies need duplicate trials, perhaps focused on insights and interpretations potentially. The present findings may therefore provide valuable information that contributes to the design of water-based exercise programs for rehabilitation.

## Figures and Tables

**Figure 1 ijerph-20-01854-f001:**
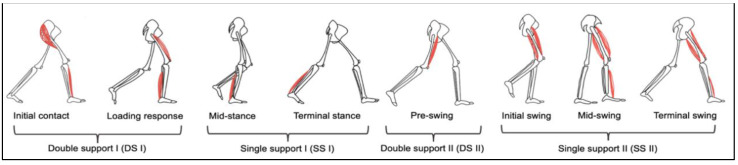
Illustration of a complete gait cycle with the right leg’s events, separating the eight gait phases: Initial contact, loading response, mid-stance, terminal stance, pre-swing, initial swing, mid-swing, and terminal swing.

**Figure 2 ijerph-20-01854-f002:**
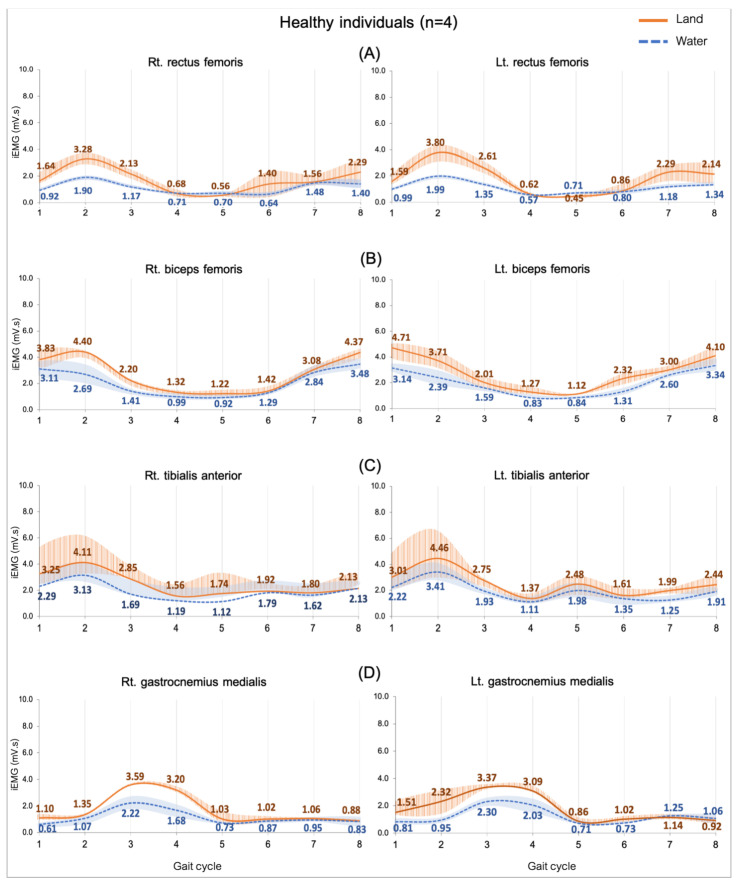
EMG activity while walking on land and in water in healthy individuals (n = 4). Rectus femoris, RF (**A**), biceps femoris, BF (**B**), tibialis anterior, TA (**C**), and gastrocnemius medialis, GM (**D**). Gait phases: 1 = initial contact, 2 = loading response, 3 = mid stance, 4 = terminal stance, 5 = pre swing, 6 = initial swing, 7 = mid swing, and 8 = terminal swing.

**Figure 3 ijerph-20-01854-f003:**
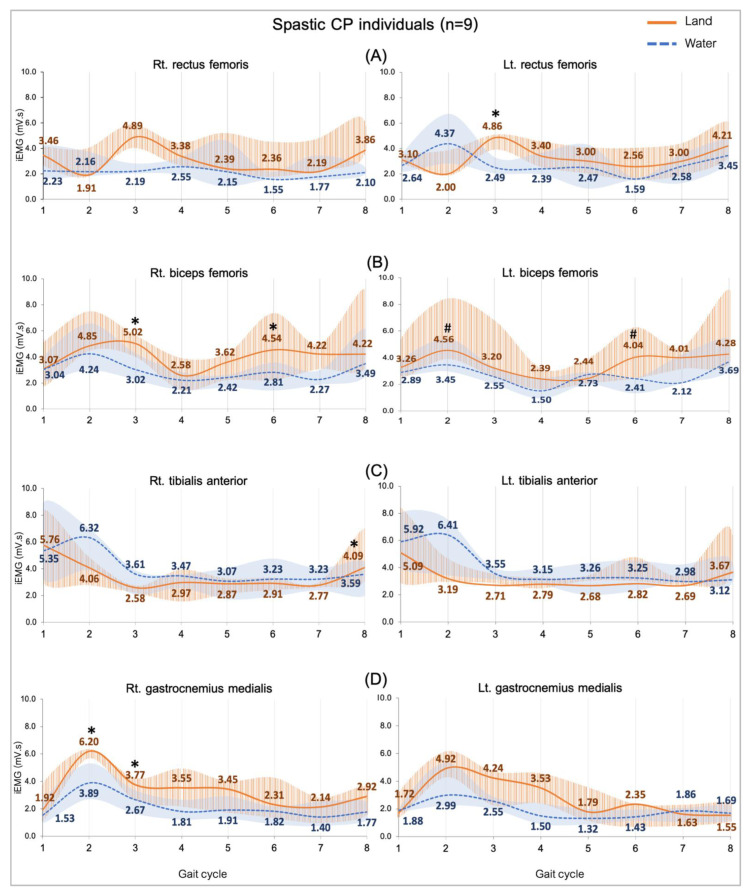
EMG activity while walking on land and in water in the spastic CP individuals (n = 9). Rectus femoris, RF (**A**), biceps femoris, BF (**B**), tibialis anterior, TA (**C**), and gastrocnemius medialis, GM (**D**). Gait phases: 1 = initial contact, 2 = loading response, 3 = mid stance, 4 = terminal stance, 5 = pre swing, 6 = initial swing, 7 = mid swing and 8 = terminal swing. * and # are significant median differences between walking on land and in water at *p* < 0.05 and *p* < 0.01.

**Figure 4 ijerph-20-01854-f004:**
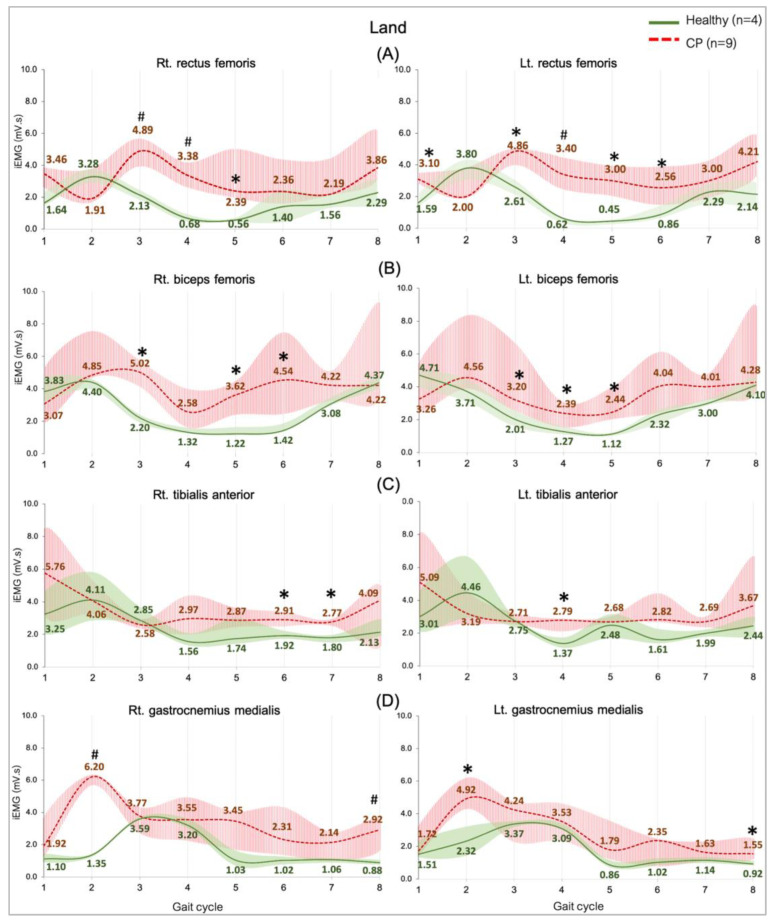
EMG activity during walking on land in the healthy (n = 4) and CP individuals (n = 9). Rectus femoris, RF (**A**), biceps femoris, BF (**B**), tibialis anterior, TA (**C**), and gastrocnemius medialis, GM (**D**). Gait phases: 1 = initial contact, 2 = loading response, 3 = mid stance, 4 = terminal stance, 5 = pre swing, 6 = initial swing, 7 = mid swing, and 8 = terminal swing. * and # are significant median differences between walking on land and in water at *p* < 0.05 and *p* < 0.01.

**Figure 5 ijerph-20-01854-f005:**
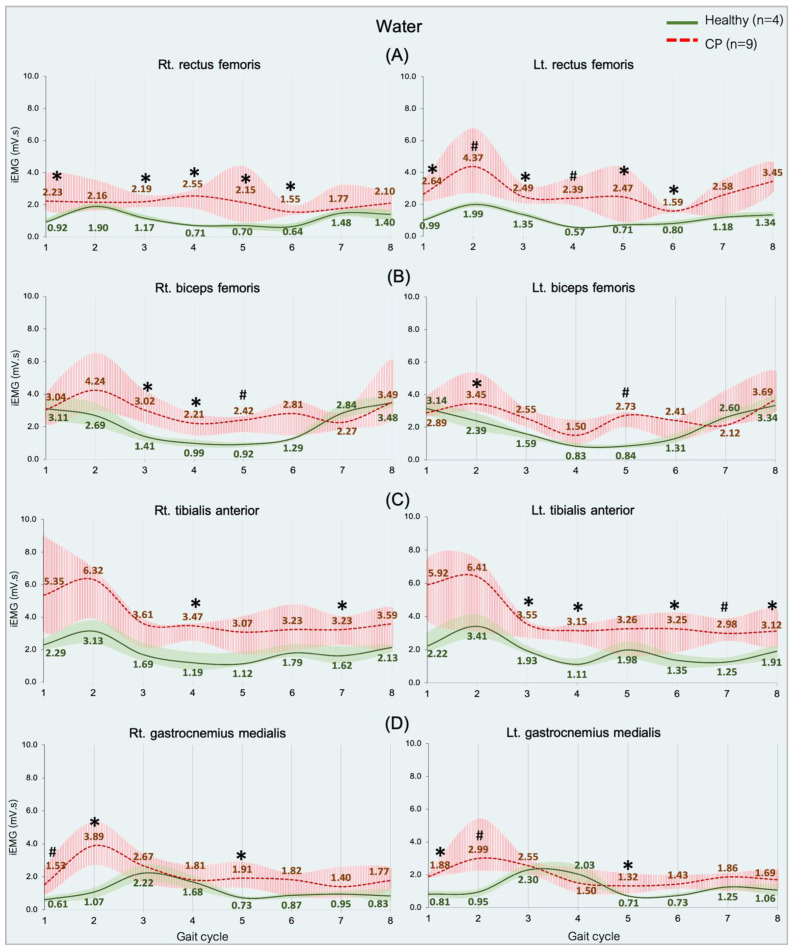
EMG activity during walking in water in the healthy (n = 4) and CP individuals (n = 9). Rectus femoris, RF (**A**), biceps femoris, BF (**B**), tibialis anterior, TA (**C**), and gastrocnemius medialis, GM (**D**). Gait phases: 1 = initial contact, 2 = loading response, 3 = mid stance, 4 = terminal stance, 5 = pre swing, 6 = initial swing, 7 = mid swing, and 8 = terminal swing. * and # are significant median differences between walking on land and in water at *p* < 0.05 and *p* < 0.01.

**Figure 6 ijerph-20-01854-f006:**
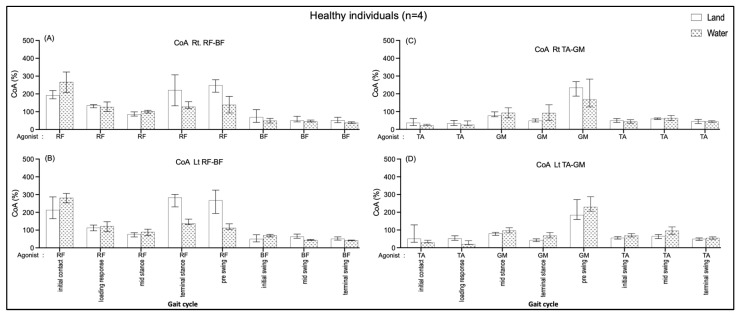
Co-activation index (CoA) of right RF-BF (**A**), left RF-BF (**B**), right TA-GM (**C**), and left TA-GM (**D**) during walking in water and on land in healthy individuals.

**Figure 7 ijerph-20-01854-f007:**
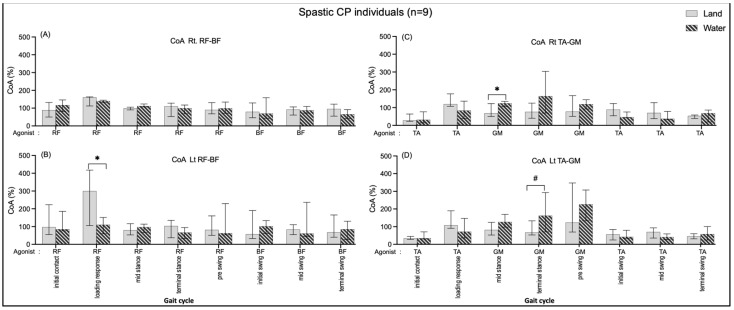
Co-activation index (CoA) of right RF-BF (**A**), left RF-BF (**B**), right TA-GM (**C**), and left TA-GM (**D**) during walking in water and on land in the CP individuals. * and # are significant median differences between walking on land and in water at *p* < 0.05 and *p* < 0.01.

**Figure 8 ijerph-20-01854-f008:**
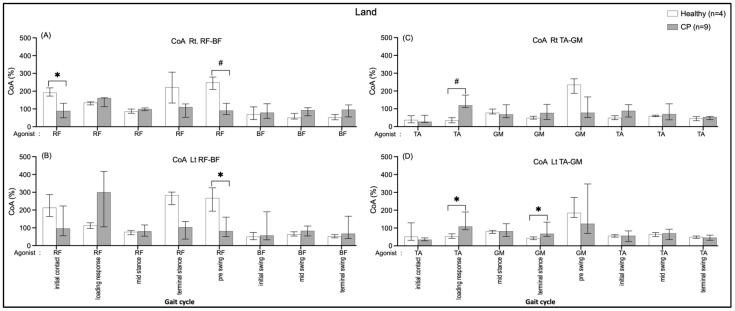
Co-activation index (CoA) of right RF-BF (**A**), left RF-BF (**B**), right TA-GM (**C**), and left TA-GM (**D**) during walking on land in healthy and CP individuals. * and # are significant median differences between walking on land and in water at *p* < 0.05 and *p* < 0.01, respectively.

**Figure 9 ijerph-20-01854-f009:**
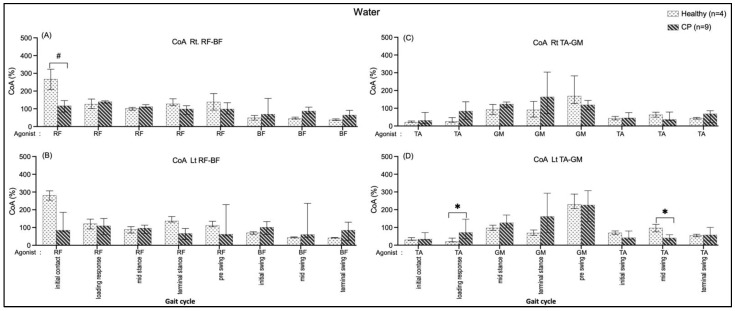
Co-activation index (CoA) of right RF-BF (**A**), left RF-BF (**B**), right TA-GM (**C**), and left TA-GM (**D**) during walking in water in healthy and CP. * and # are significant median differences between walking on land and in water at *p* < 0.05 and *p* < 0.01.

**Table 1 ijerph-20-01854-t001:** Demographics of cerebral palsy and healthy subjects.

	No.	Age	Sex	Height (m)	Weight (Kg)	BMI (Kg/m^2^)	GMFCs
Healthy	1	16	M	1.23	25.60	16.92	-
	2	15	F	1.53	38.00	16.23	-
	3	13	M	1.60	41.00	16.02	-
	4	12	M	1.61	55.00	21.22	-
Median		14.00	3M,1F	1.57	39.50	16.58	
*Q1, Q3*		12.75, 15.25		1.46, 1.60	34.90, 44.50	16.16, 18.00	
**CP**	1	16	M	1.55	44.68	18.60	I
	2	15	M	1.36	25.00	13.52	I
	3	13	M	1.34	32.00	17.82	I
	4	12	M	1.29	21.50	12.92	I
	5	16	M	1.45	26.70	12.70	I
	6	15	M	1.42	37.12	18.41	I
	7	16	M	1.50	36.70	16.31	I
	8	17	F	1.53	45.00	19.22	II
	9	13	M	1.38	41.00	21.53	II
Median		15.00	8M,1F	1.42	36.7	17.82	
*Q1, Q3*		13.00, 16.00		1.36, 1.50	26.7, 40.00	13.52, 18.60	

BMI; body mass index, GMFCs; gross motor function classification system.

**Table 2 ijerph-20-01854-t002:** Spatiotemporal parameters of a gait cycle in CP and healthy subjects.

	Healthy (n = 4)	CP (n = 9)	Between Gr. *p*-Value
Walking speed (m/s)			
• Land	1.15 [1.04,1.27]	0.83 [0.69,0.91]	0.02 *
• Water	1.07 [0.96,1.15]	0.61 [0.49,0.74]	0.009 ^#^
Within-gr. *p*-value	0.07	0.008 ^#^	
Rt. Cadence (step/s)			
• Land	1.82 [1.65,2.06]	1.98 [1.70,2.09]	1.00
• Water	2.17 [1.99,2.30]	1.86 [1.59,2.15]	0.28
Within-gr. *p*-value	0.27	0.59	
Lt. Cadence (step/s)			
• Land	1.79 [1.60,1.92]	1.92 [1.56,1.99]	0.54
• Water	2.09 [2.02,2.15]	1.88 [1.70,2.03]	0.22
Within-gr. *p*-value	0.07	0.59	
Rt. Gait cycle time (s)			
• Land	1.00 [0.97,1.04]	1.06 [1.05,1.10]	0.36
• Water	1.60 [1.48,1.74]	1.55 [1.30,1.57]	0.28
Within-gr. *p*-value	0.07	0.008 *	
Lt. Gait cycle time (s)			
• Land	1.06 [1.00,1.09]	1.18 [1.11,1.21]	0.44
• Water	1.59 [1.52,1.69]	1.55 [1.30,1.57]	0.22
Within-gr. *p*-value	0.07	0.015 *	
Rt. % Stance phase			
• Land	64.33 [63.35,65.23]	67.70 [67.42,69.45]	0.03 *
• Water	62.25 [61.25,62.91]	65.33 [62.42,67.45]	0.12
Within-gr. *p*-value	0.07	0.03 *	
Lt. % Stance phase			
• Land	63.38 [62.44,64.26]	67.90 [66.73,68.35]	0.02 *
• Water	62.88 [62.54,63.33]	65.34 [63.09,65.71]	0.19
Within-gr. *p*-value	1.00	0.011 *	
Rt. % Swing phase			
• Land	35.67 [34.77,36.66]	32.30 [30.55,32.58]	0.03 *
• Water	37.81 [37.09,38.75]	34.67 [32.55,37.58]	0.12
Within-gr. *p*-value	0.07	0.03 *	
Lt. % Swing phase			
• Land	36.63 [35.74,37.56]	32.10 [31.66,33.27]	0.02 *
• Water	37.12 [36.67,37.46]	34.66 [34.29,36.91]	0.19
Within-gr. *p*-value	0.46	0.011 *	

Data are presented as median [Q1, Q3]. Statistical analysis by Wilcoxon signed-rank test for comparisons within group and Mann–Whitney U test for comparisons between groups. *, ^#^ are significant differences of *p* < 0.05 and *p* < 0.01.

## Data Availability

The data is unavailable due to privacy or ethical restrictions.

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
