# Peer review of "Muscle Activity and Co-Activation of Gait Cycle during Walking in Water and on Land in People with Spastic Cerebral Palsy"

_ijerph, 2023, doi:10.3390/ijerph20031854_

Round 1
Reviewer 1 Report
Proposed paper for publication gives precious insight into muscle activity and co-activation of gait cycle during walking in water.
There are some minor changes to be made through the text. Authors use phrase "healthy" subjects in comparison to subjects with CP. CP is a group of disorders that affect a person's ability to move and maintain balance and posture and can't be considered as an ilness. Please correct through the text.
Demographics of the subjects are properly described in chapter Results. They subject sample is usually presnted in chapter Material and Methods in paragraph Subjects. Please correct.
Results are presented in clear and proper way.
In discussion authors conclude that the present findings provide valuable information that contributes to the designof water-based exercise programs for rehabilitation. However, in the discussion could be better explained how findings of the study are important and how water rehabilitiaion programmes based on walking could contribute to better daily live of persons with CP. Please explain better.
Author Response
Response to Reviewer 1 Comments
Dear Reviewer 1,
I would like to thank you for your suggestion. We carefully revised the manuscript following below.
Comments and Suggestions for Authors
Proposed paper for publication gives precious insight into muscle activity and co-activation of gait cycle during walking in water.
Point 1: There are some minor changes to be made through the text. Authors use phrase "healthy" subjects in comparison to subjects with CP. CP is a group of disorders that affect a person's ability to move and maintain balance and posture and can't be considered as an ilness. Please correct through the text.
Response 1: Revised all words of “patients with CP” to “subjects with CP or people with CP”
Point 2: Demographics of the subjects are properly described in chapter Results. They subject sample is usually presented in chapter Material and Methods in paragraph Subjects. Please correct.
Response 2: Revised describing the demographics of the subjects in the part of Results (line 171-176) and the subject sample in the part of Subjects (line 66-69,81-86)
Results are presented in clear and proper way.
Point 3: In discussion authors conclude that the present findings provide valuable information that contributes to the design of water-based exercise programs for rehabilitation. However, in the discussion could be better explained how findings of the study are important and how water rehabilitiaion programmes based on walking could contribute to better daily live of persons with CP. Please explain better.
Response 3: added the explanation (line 414-420)
“The present findings provide valuable information that contributes to the design of water-based exercise programs for rehabilitation. That is, walking in water could decrease muscle activity in spastic lower limbs, which may generate easier stepping and lead to success in walking. In addition, walking in water may promote normalizing muscle tone in preparation for physical functional training. Finally, the improved walking ability could enhance daily living for persons with CP. This mention is likely suitable for persons with spastic CP.”
Best regards,
PP

Reviewer 2 Report
Congratulations to the authors for the development of this research titled “Muscle activity and co-activation of gait cycle during walking in water and on land in people with spastic cerebral palsy”. The manuscript provides information about the differences of muscle activity and co-activation index of the rectus femoris, biceps femoris, gastrocnemius medialis and tibialis anterior during walking on land and in water in healthy adolescents compared with those with spastic diplegia cerebral palsy adolescents.
Some minor suggestion are recommended, to improve the quality of the manuscript.
Abstract:
It is very well written and includes most of the information of the investigation.
Line 14: delete the word "the"
Line 24: ..... the all muscles ..... Please correct
Introduction:
Well structured and justifies the need for the work.
Line 31: please take care of the punctuation
Line 40: Take care of the punctuation
Material and Methods:
Very well written and structured.
Line 103: (data collection), Please correct
Line 141: Take care of the punctuation
Line 150: Terminal, Please correct
Results:
The Figures and Tables are clear and very well presented.
Line 197: .... was significantly....
The Figure 9. should go to the results section.
Discussion
This section is very well developed.
Study limitations is necessary and most of the information are included.
Line 285: Delete the word "the"
Author Response
Dear Reviewer 2,
I would like to thank you for your suggestion. We carefully revised the manuscript following below.
Response to Reviewer 2 Comments
Congratulations to the authors for the development of this research titled “Muscle activity and co-activation of gait cycle during walking in water and on land in people with spastic cerebral palsy”. The manuscript provides information about the differences of muscle activity and co-activation index of the rectus femoris, biceps femoris, gastrocnemius medialis and tibialis anterior during walking on land and in water in healthy adolescents compared with those with spastic diplegia cerebral palsy adolescents.
Some minor suggestion are recommended, to improve the quality of the manuscript.
Abstract:
It is very well written and includes most of the information of the investigation.
Point 1: Line 14: delete the word "the"
Response 1: delated (line 14)
Point 2: Line 24: ..... the all muscles ..... Please correct
Response 2: revised (line 24)
Introduction:
Well structured and justifies the need for the work.
Point 3: Line 31: please take care of the punctuation
Response 3: added punctuation “ , ” (line 31 )
Point 4: Line 40: Take care of the punctuation
Response 4: added punctuation “ , ” (line 40 )
Material and Methods:
Very well written and structured.
Point 5: Line 103: (data collection), Please correct
Response 5: revised (line 103)
Point 6: Line 141: Take care of the punctuation
Response 6: revised (line 140)
Point 7: Line 150: Terminal, Please correct
Response 7: revised (line 149)
Results:
The Figures and Tables are clear and very well presented.
Point 8: Line 197: .... was significantly....
Response 8: added (line 203)
Point 9: The Figure 9. should go to the results section.
Response 9: revised (line 288)
Discussion
This section is very well developed.
Study limitations is necessary and most of the information are included.
Point 10: Line 285: Delete the word "the"
Response 10: delated (line 294)
Best regards,
PP

Reviewer 3 Report
It deals with an exciting and relevant topic. Some typographical and grammar errors should be corrected. Kindly explain the comparison of the group as the sample size is small and the number of subjects differs in both groups. Needs to add new references.
Author Response
Dear Reviewer 3,
I would like to thank you for your suggestion. We carefully revised the manuscript following below.
Response to Reviewer 3 Comments
Comments and Suggestions for Authors
It deals with an exciting and relevant topic.
Point 1: Some typographical and grammar errors should be corrected.
Response 1: recheck all already
Point 2: Kindly explain the comparison of the group as the sample size is small and the number of subjects differs in both groups. Needs to add new references.
Response 2: added the paragraph (line 421-427)
“Limitations of this study were a few. A small sample size in healthy humans could limit the statistical power to expose such a small effect, possibly resulting in a type II error. Because of the study was conducted during the covid-19 pandemic situation which caused an incomplete sample number. Nevertheless, the healthy subjects' demographic characteristics (age, height and weight) similar to the spastic CP subjects, and no significant difference in spatiotemporal parameters (cadence and gait cycle time). Consideration of the muscle activity (iEMG) variable seems a small variation in the healthy group.”
Best regards,
PP
